# Reverse Torque Value of Angulated Screw Channel Abutment before and after Cyclic Loading: An In Vitro Study

**DOI:** 10.3390/jfb14030124

**Published:** 2023-02-24

**Authors:** Yu-Hsuan Chen, Yu-Ling Wu, Hung-Shyong Chen, Ching-Ping Lin, Aaron Yu-Jen Wu

**Affiliations:** 1Department of Dentistry, Kaohsiung Chang Gung Memorial Hospital and Chang Gung University College of Medicine, Kaohsiung 833, Taiwan; 2Department of Mechanical Engineering, Cheng Shiu University, Kaohsiung 833, Taiwan; 3Center for Environmental Toxin and Emerging-Contaminant Research, Cheng Shiu University, Kaohsiung 833, Taiwan

**Keywords:** implant, angulated screw-retained crown, angulated screw channel, reverse torque value, cyclic loading, mechanical stresses

## Abstract

This in vitro experiment aimed to understand the difference in preload acting on an abutment screw under different angles of angulated screw-retained crown and the performance after cyclic loading. In total, thirty implants with angulated screw channel (ASC) abutments were divided into two parts. The first part consisted of three groups: a 0° access channel with a zirconia crown (ASC-0) (*n* = 5), a 15° access channel with a specially designed zirconia crown (sASC-15) (*n* = 5), and a 25° access channel with a specially designed zirconia crown (sASC-25) (*n* = 5). The reverse torque value (RTV) was measured at 0° for each specimen. The second part consisted of three groups: a 0° access channel with a zirconia crown (ASC-0) (*n* = 5); a 15° access channel with a zirconia crown (ASC-15) (*n* = 5), and a 25° access channel with a zirconia crown (ASC-25) (*n* = 5). The manufacturer’s recommended torque was applied to each specimen, and baseline RTV was measured before cyclic loading. Each ASC implant assembly was cyclically loaded at 0 to 40 N with 1 million cycles at 10 Hz. RTV was measured after cyclic loading. Kruskal–Wallis test and Jonckheere–Terpstra test were used for statistical analysis. All specimens were examined under a digital microscope and scanning electron microscope (SEM) to observe the wear of the screw head before and after the whole experiment. A significant difference in the different percentages of straight RTV (sRTV) between the three groups was found (*p* = 0.027). The angle of ASC to the different percentages of sRTV showed a significant linear trend (*p* = 0.003). No significant differences were found in RTV difference after cyclic loading among the ASC-0, ASC-15, and ASC-25 groups (*p* = 0.212). The ASC-25 group had the most serious degree of wear based on a digital microscope and SEM examination. The ASC angle will affect the actual preload acting on a screw: the larger the ASC angle, the smaller the preload. The performance of the angled ASC groups in RTV difference was comparable to that of 0° ASC after cyclic loading.

## 1. Introduction

The reconstruction of edentulous areas with implants is a highly predictable approach with high success and survival rates [1,2]. A cohort study by French et al. showed that the 10-year implant survival rate was 96.8% [3]. However, when restoring the aesthetic area of the anterior teeth using implants, considering the hard, and soft tissues, the retained types of implant crowns are also critical.

Traditionally, a cement-retained implant prosthesis is often used in the aesthetic area due to anatomical conditions, better mechanical behavior, and fewer mechanical complication such as screw loosening [4,5,6], but residual cement may cause peri-implantitis [7,8]. One way to avoid this problem might be to use a screw-retained crown (SRC) instead. The advantage of SRCs is that they are easy to retrieve, and they avoid complications caused by residual cement. However, if a screw-retained implant prosthesis is used, the screw hole may be on the buccal side, and aesthetics will be affected. To solve this problem, there are specially designed implant abutments and hexalobular drivers on the market [9]. The special design allows the screw channel of the abutment to be in an ideal position, avoiding the buccal side screw holes. However, in clinical use, there may be concerns that an angle screw channel is easier to loosen than a straight screw-retained abutment, even though some manufacturers claim that the torque of the tightening screw at different angles will not be different. Several in vitro studies compared the performance of angulated screw-retained crowns with that of straight screw-retained crowns, and experiments found no difference in resistance in the loosening of angulated screw channel (ASC) retained crowns and straight screw-retained crowns under functional loading [10,11]. The ASC designed by Nobel Biocare can place the screw access hole anywhere between 0° and 25° within a 360° radius. Most ASC-retained crowns on the market are fixed on the Ti-base by cement. However, ASC crowns and Ti-base rely on fitted friction (Figure 1) [12].

The most common complications of implant-supported prostheses are screw loosening, screw fracture, and implant fracture. Among them, screw loosening is the most common with a rate of 2.1% [5]; it can be caused by several factors, including screw settling, and excessive functional loading. Screw settling is caused by the micro-roughness loss of the screw thread and internal surface during loading [13]. Functional loading may cause screw loosening if it is equal to or greater than the preload of the screw [14]. The torque applied to the screw clinically can provide a clamping force to the screw and abutment. When the torque falls below the manufacturer’s recommended value, the clamping force may drop, and the screw will loosen [15]. Many studies used a reverse torque value (RTV) to understand the degree of screw tightening [10,11,15,16]. Some studies pointed out that an ASC can cause RTV to drop after cyclic loading [10,12]; however, some studies showed no significant difference in RTV between different angles after cyclic loading [10,11]. However, minimal research has been published to evaluate the ability of ASC to resist loosening at different angles of screw access, which can make dentists hesitate in using ASC in clinical practice. Therefore, the purpose of this in vitro study was to compare the difference in RTV at different angles of ASC before and after simulated functional loading and understand whether the different angles affect clinical performance. This experiment was divided into two parts: the first part involved using specially designed zirconia crowns to understand the preload difference acting on an abutment screw under different ASC angles and the second part comprised comparing the changes in RTV at different ASC angles after cyclic loading and observing whether a screw head was worn under a digital microscope and scanning electron microscope (SEM).

## 2. Materials and Methods

In total, 30 implants (Nobel Replace Conical Connection PMC Ø = 4.3 mm × 10 mm, Nobel Biocare, Goteborg, Sweden and Nobel ASC abutment (Nobel Biocare, Goteborg, Sweden) with a full zirconia crown (NobelProcera; Nobel Biocare, Goteborg, Sweden). To be closer to the clinical situation and the standardized reproduction of all the zirconia crowns, a wax pattern replicating a maxillary right central incisor was scanned (KaVo LS3; KaVo) to a standard tessellation language file.

### 2.1. Straight Reverse Torque Value

To understand whether the actual preload acting on an abutment screw will be affected by the ASC angle, 15 implants with a special design of zirconia crown with customized access screw hole (Figure 2A–D) were divided into three angulation (0°, 15°, and 25°) groups. The crown of the 0° ASC (ASC-0) group only allowed the screwdriver to input a torque value at a 0° angle (*n* = 5). The crown of the 15° ASC (sASC-15) group allowed the screwdriver to input a torque value at either 0° or 15° angle (*n* = 5). The crown of the 25° ASC (sASC-25) group allowed the screwdriver to input a torque value at either 0° or 25° angle (*n* = 5); the implant was fixed to a specially designed metal jig using angle vice (Figure 3A,B), which guided the screwdriver to input torque values at specific angulations using a digital torque meter (Tohnichi BTGE50CN-G) (Figure 3C) to tighten the crown and Ti-base to the implant at 0°, 15°, and 25° according to the torque value (35 Ncm) recommended by the manufacturer. After the ASC screw was tightened to 35 Ncm, a digital torque meter was used to tighten the screw to 35 Ncm again after 10 min. For each group, RTV was recorded at 0° via a specially designed zirconia crown using a digital torque meter. This step was repeated thrice for each specimen, and the average of these three data was defined as sRTV. The tightening and loosening processes were repeated for all specimens using the same screws. The percentage of difference in sRTV was calculated with the following equation: (sRTV − 35)/35 × 100%.

### 2.2. Angulated Reverse Torque Value and Cyclic Loading

Fifteen implants were divided into three groups according to different ASC angles. The ASC-0, ASC-15, and ASC-25 groups had a 0°, 15°, and 25° ASC abutment with a screw-retained zirconia crown, respectively (all, *n* = 5; Figure 2E).

After fixing the implant to the metal jig, the crown and Ti-base were tightened to the implant using a digital torque meter with a torque value of 35 Ncm. After 10 min, the ASC screw was tightened to 35 Ncm again, and RTV was recorded using the digital torque meter. These steps were repeated thrice for each specimen with the same screw, and the average RTV was calculated; this was defined as the baseline RTV (bRTV). 

To understand the difference in RTV changes at different ASC angles after cyclic loading, the samples were fixed in a universal testing machine (Instron Electropuls E-3000; Instron) (Figure 4A). The metal jig was clamped after aligning it with the vise on the universal testing machine using the parallel block to standardize the position of the load and ensure the repeatability and reproducibility of the experiment (Figure 4B). All specimens with the same screws were tightened to 35 Ncm, and a metal rod with a 6 mm diameter was used to apply a force of 0–40 N at a 30° angle to the implant axis (Figure 4C) [10,17]. In total, 1 million cycles at a frequency of 10 Hz were used to simulate the situation, where the maxillary central incisor was functionally loaded for one year [10,11,16]. After 10 min, the screw was re-tightened to 35 Ncm to compensate for the loss of initial preload due to the settling effect [10,13]. When the cyclic loading was completed, the removal torque value (cRTV) was recorded with the digital torque meter. The difference in RTV was calculated with the following equation: cRTV-bRTV.

### 2.3. Microscope and SEM Examination

Prior to the experiment, the side of the screw head in each group of ASC screws was marked to obtain images before and after cycling at exactly the same position using a digital microscope (Keyence VHX-900F) and SEM (JSM-6360, JEOL) with a 20 kV accelerating voltage at ×100 magnification.

After the whole experiment, the specimens were untightened using a torque device. The ASC screw hex was inspected under a digital microscope at ×100 magnification and SEM at ×100 magnification to observe whether the screw hex was worn or not. Figure 5 shows the outline of the experiments.

#### Statistical Analysis

Kruskal–Wallis test was used to analyze the change percentage in sRTV at different ASC angles and torque value loss after cyclic loading. Jonckheere–Terpstra test was used to analyze whether there is a linear trend between different groups. The statistical software used was SPSS (IBM SPSS Statistics v28; IBM Corp, Armonk, NY, USA). 

## 3. Results

### 3.1. Straight Reverse Torque Value Percentage Difference 

The sRTV of all samples was lower than the input torque value, and there was a significant difference among sASC-0, sASC-15, and sASC-25 (*p* = 0.027) (Table 1). Table 1 and Figure 6 show the reduction percentage difference for each group. In addition, there was a significant linear trend between the ASC angles and sRTV difference percentages (*p* = 0.003).

### 3.2. Angulated Reverse Torque Value and RTV Difference after Cyclic Loading

The mean bRTV values of the ASC-0, ASC-15, and ASC-25 groups were 28.84 ± 0.91, 28.87 ± 1.43, and 27.14 ± 0.80 Ncm, respectively (Table 2). The RTV of all samples was less than the manufacturer’s recommended input torque value.

After cyclic loading, the RTVs of all samples were lower than the input torque value, and the RTV differences of the ASC-0, ASC-15, and ASC-25 groups were −2.41 ± 1.68, −2.70 ± 1.44, and −1.38 ± 1.40 Ncm, respectively. However, there was no significant difference between the three groups (*p* = 0.212; Figure 7; Table 2).

### 3.3. Microscope and SEM Examination

No crown mobility, crown fractures, screw fractures, or implant fractures were noted after the whole experiment. Based on digital microscope investigation, the inner wall of the screw head had obvious irregular depressions, and the hexagon was also worn to a flat obtuse angle. This wear condition was found in all used ASC screws, and it became more severe as the ASC angle increased (Figure 8). Comparing the SEM micrographs before and after the experiment, more wear and damage on the screw head were recorded, and pit depressions and obvious surface bow-shaped indentations were found under SEM. The degree of wear in the ASC-25 and ASC-15 groups was greater than that in the ASC-0 group, and the ASC-25 group had the most serious wear (Figure 9).

## 4. Discussion

Even though a higher risk of mechanical issues is reported in the literature, the screw-retained prosthesis is a reliable treatment and may have some biological advantages over the cement-retained prosthesis [18,19,20,21,22,23]. Although angulated SRCs can make it possible for dentists to use screw-retained crowns under the condition that only cement-retained crowns can be used, because of a lack of evidence and long-term clinical outcome data, dentists will be hesitant in using this unfamiliar component. Screw loosening is one of the most common prosthetic complications, especially in single-tooth replacement with implants [8]. One reason for loose screws is an insufficient preload. When the external load is beyond the preload, the abutment screw connection loses stability, affecting the vibration, and micromovement of the interfaces, which causes screw loosening [20,21,22,23]. An angulated SRC is usually used when the implant is in a suboptimal position; this may cause the implant abutment screw to receive larger lateral forces when functionally loaded, which may increase the chance of screw loosening. The preload can be defined as tension on the screw when a torque is applied to the screw head, preventing its components from separating [24]. Only 10% of the initial torque is converted into preload, while the remaining 90% is used to overcome the friction between surface irregularities [15,25]. The applied torque creates a force called preload within the screw. The preload applied to the screw provides a clamping force between the screw and the Ti-base against external forces that cause the screw Ti-base to separate; the clamping force is proportional to the insertion torque [15,26]. However, the preload force should not be too large and should be within the elastic limit to prevent the screw from yielding or breaking. Previous studies showed that the ideal preload is 60–75% of the elastic limit of the material used to make abutment screws [27,28,29]. Each manufacturer has its recommended torque value; for ASC, it is 35 Ncm. 

There are two main methods to measure preload at present. One is using strain gauges to measure the tension when attaching the abutment screw to the implant fixture. However, it is time-consuming and effort-intensive [30,31,32]. Another method is the measurement of RTV. Most studies pointed out that the tightening torque value is proportional to the preload; therefore, the current research uses the measurement of RTV to predict the changes in preload [33]. Our bRTV results show that the ASC at three different angles decreased compared to the input torque value. The difference percentage in RTV of the ASC-0 group was 17.6%, which is similar to the findings of Al-Otaibi et al., which was 16.6% when using a 0° access channel abutment (Gold-Adapt, Nobel Biocare) [34]. Another study compared the RTV of ASC-0 and Gold-Adapt 0° access channel abutment, and the median RTV percentage differences were 34.6% and 49.4%, respectively [8]. A study by Pournasiri et al. showed that the mean RTV percentage difference was 12.2% when using a Straumann implant system [35]. The decrease in torque value may be related to the settling effect [36] and friction. According to Dixon et al., about 2–10% of the initial preload will be lost due to the settling effect [37]. Bakaeen et al. suggested waiting for 10 min after the initial torque application to retorque the abutment screw so as to compensate for the preload lost due to the settling effect [38]. 

In theory, when a screwdriver is aligned along the long axis of the abutment screw, 100% of the applied force is transferred to the screw [23]. Opler et al. used a dynamic abutment system to measure the RTV at different angles, and the result in the 28° group was significantly lower than that in the 0° group [39]. To verify whether the torque value applied to the screw under this angle was different depending on the ASC angle, the ASC crowns were designed to allow the hexalobular screwdriver to apply torque at 0–15° or 0–25° at the same time. Experimental results showed that the removal torque value of the sASC-25 and sASC-15 groups when loosening at 0° was significantly smaller than that of the ASC-0 group. This may be due to the decrease in the torque value transferring to the screw when the applied torque was not aligned along the long axis of the abutment screw. The current study also showed a linear trend between the degree of ASC and the percentage difference of sRTV; thus, the greater the ASC angle, the greater the value of the RTV drop. This means that the larger the angle, the smaller the preload of the screw. It was speculated that the actual preload acting on the screw may be one of the reasons for the manufacturer-recommended ASC angle limit, which was set at 25°. Hu et al. used an S-Link abutment to tighten the screws at 0°, 10°, and 20° to the manufacturer-recommended 35 Ncm; then, the screws were loosened in the 0° direction. Their results showed that RTV was less than the target torque value from all angulation groups, which dropped 10% below the target torque value set by the manufacturer. The above study found that the mean RTV of the 10° group was the highest, while the 20° group had the lowest mean RTV [26]. According to the mechanics, applying torque at an angle to the screw will lead to a decrease in preload as the angle increases. Moreover, according to a study by Siamos et al., increasing the torque value for abutment screws can be beneficial for abutment-implant stability and to decrease screw loosening [15]. 

Many studies pointed out that RTV will decrease significantly after cyclic loading, the possible reason being that loading at an angle will generate a force of screw loosening [40,41]. Hein et al. conducted an in vitro study and showed that when comparing indexed and nonindexed with straight, 17°, and 30° abutment, RTV dropped after cyclic loading among all groups [42]. A study by Ha et al. compared straight and angled abutments after cyclic loading and found similar results [16]. According to a study by Goldberg et al., for a dynamic abutment with a similar concept to an ASC, after cyclic loading, there was no significant difference in RTV from different angles, and the angle of the abutment had no significant effect on screw removal torque values [10]. Swamidass et al. also showed that there was no significant difference in RTV between the 0° ASC and 20° ASC groups after cyclic loading [11]. The results of the present study also showed that there was no significant difference in RTV decline between 0° ASC and ASC at different angles after cyclic loading. However, the ASC-25 group had a smaller decrease in RTV after cyclic loading. According to the experimental results of the first part of the experiment, the preload of the ASC-25 group was significantly smaller than that of the ASC-0 group, so it can be speculated that if the screws were under different preloads during cyclic loading, the degree of screw loosening may be different. Thus, the reduction in the RTV of ASC after cyclic loading will not be different due to ASC angles, so even if an ASC with a larger angle is used, it should be able to obtain an ideal clinical performance. Therefore, ASC may be a reliable choice when clinically encountering an inappropriate implant position but wishing to use screw-retained crowns. 

No crown mobility or fracture occurred in all samples, but wear was found on the inner wall and hex of the ASC screwhead after repeated tightened and loosening (Figure 6 and Figure 7). Future experiments can use the 3D scanner to quantify the amount of wear on abutment and implant components and explore their relationship with RTV changes. In addition, it is crucial to explore whether different angles of ASC can affect the amount of wear on components. The present study has some limitations. As per the ISO 14801:2007 protocol for the cycling load test, the standardized shape of the crown is hemispheric; however, this shape is too idealized to be different from the actual crown shape. Therefore, a full zirconia crown with the shape of the maxillary right central incisor was used in this study to simulate the clinical loading. However, different crown morphologies may lead to different results after cyclic loading. In addition, the clamping device or embedding used to fix the specimen may affected the experiment results. Future experiments may consider using other clamping devices or embedding materials, such as jawbone or other resins, having an elastic modulus greater than 3 GPa, and discuss the similarities and differences of the experimental results.

## 5. Conclusions

Within the limitations of the method and based on the results of this in vitro study, the following conclusions were drawn:

The ASC angle will affect the actual preload acting on the abutment screw and the larger the ASC angle, the lower the preload.The RTV of the ASC of all groups decreased after cyclic loading, and there was no significant difference between the 0° and angled groups. Thus, the performance of angled ASCs after cyclic loading was comparable to that of 0° ASC.

## Figures and Tables

**Figure 1 jfb-14-00124-f001:**
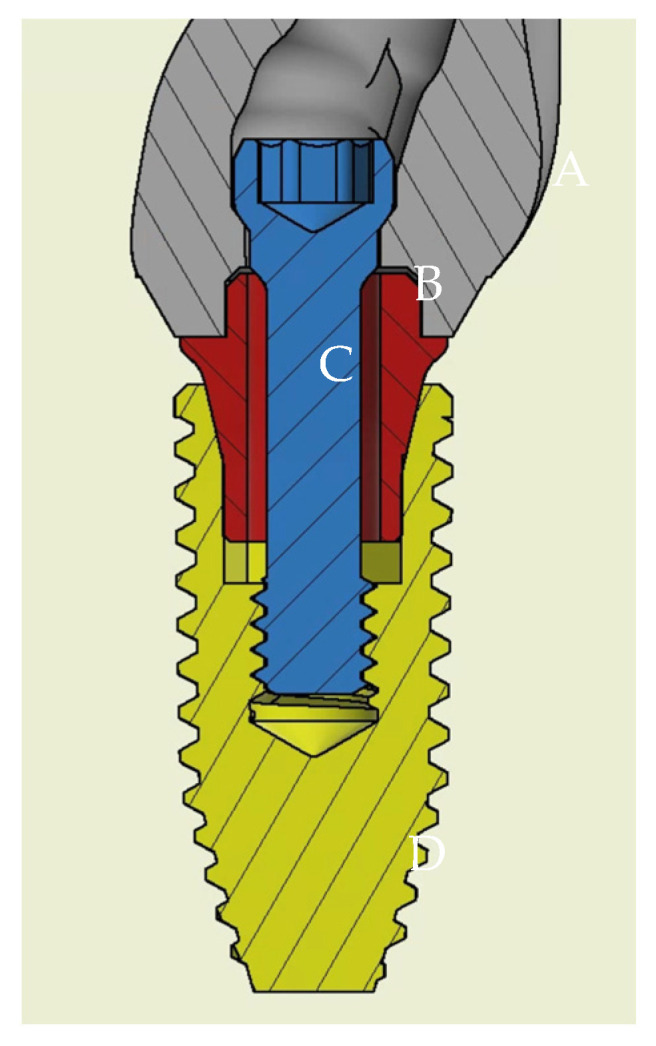
Section diagram of ASC assembly and implant. (**A**) Full zirconia crown (NobelProcera); (**B**) screw of ASC; (**C**) Ti-base of ASC; (**D**) Nobel Replace Conical Connection PMC Ø = 4.3 mm × 10 mm.

**Figure 2 jfb-14-00124-f002:**
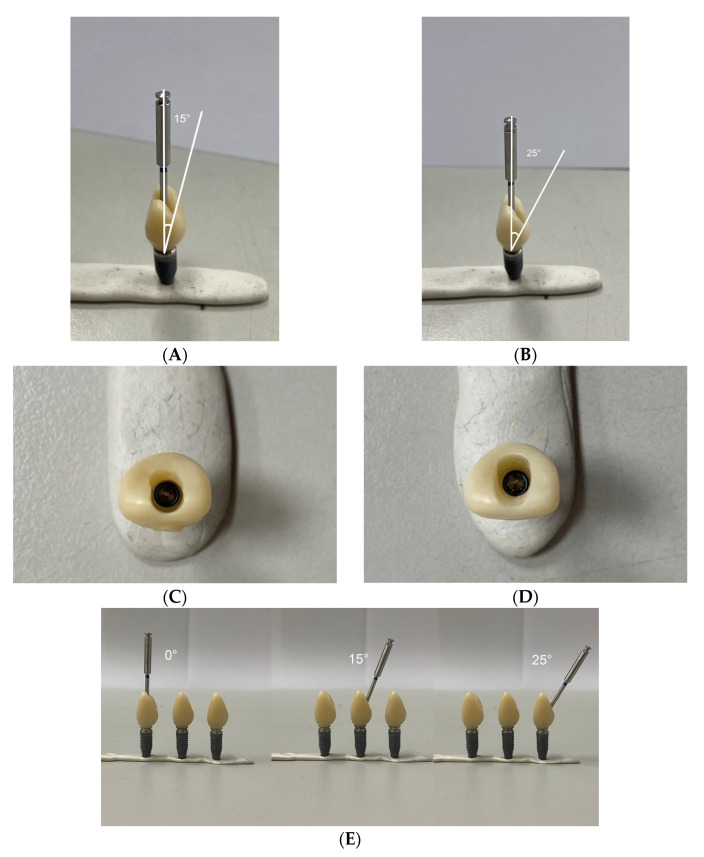
(**A**) The special design of zirconia crown allowed the Omnigrip screwdriver to apply at the angle of 15° or 0°(sASC-15). (**B**) The special design of zirconia crown allowed the Omnigrip screwdriver to apply at the angle of 25° or 0° (sASC-25). (**C**) The incisal view of sASC-15. (**D**) The incisal view of sASC-25. (**E**) The 0°,15°,25° ASC abutment with Nobel Biocare zirconia crowns (ASC-0, ASC-15, ASC-25).

**Figure 3 jfb-14-00124-f003:**
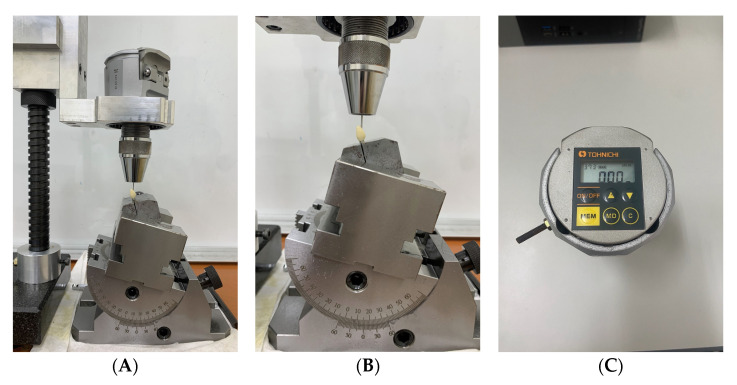
(**A**) Reverse torque values (RTV) recorded by the digital torque meter with the Omnigrip screwdriver guided by the special design metal jig with angle vice. (**B**) A specially designed metal jig that clamps the implant on a 30° axis. The angle vice guides the Omnigrip to input torque at a specific angulation. (**C**) Tohnichi BTGE50CN-G digital torque meter.

**Figure 4 jfb-14-00124-f004:**
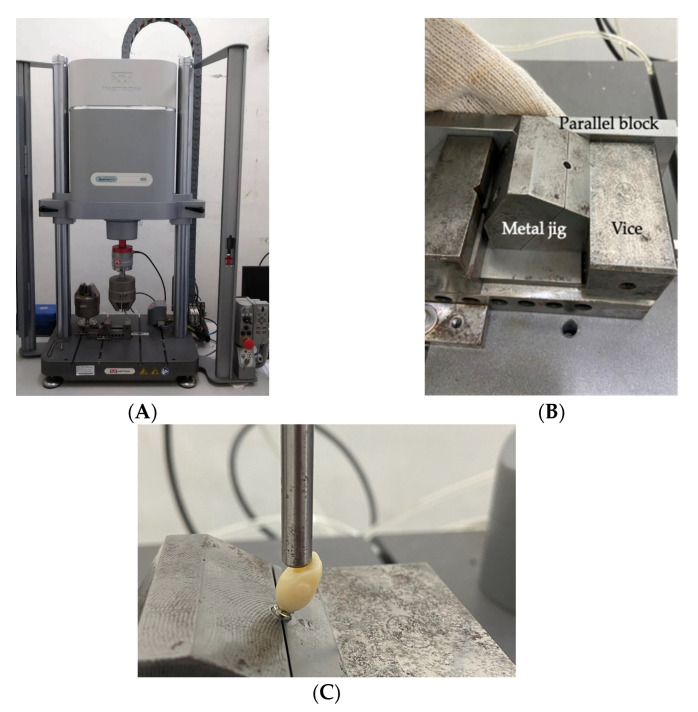
(**A**) Universal testing machine (Instron Electropuls E-3000; Instron). (**B**) The metal jig is aligned with the vise on the universal testing machine through the parallel block. (**C**) The implant was clamped at the level of apical border of machine collar; metal rod with a diameter of 6 mm for cyclic loading.

**Figure 5 jfb-14-00124-f005:**
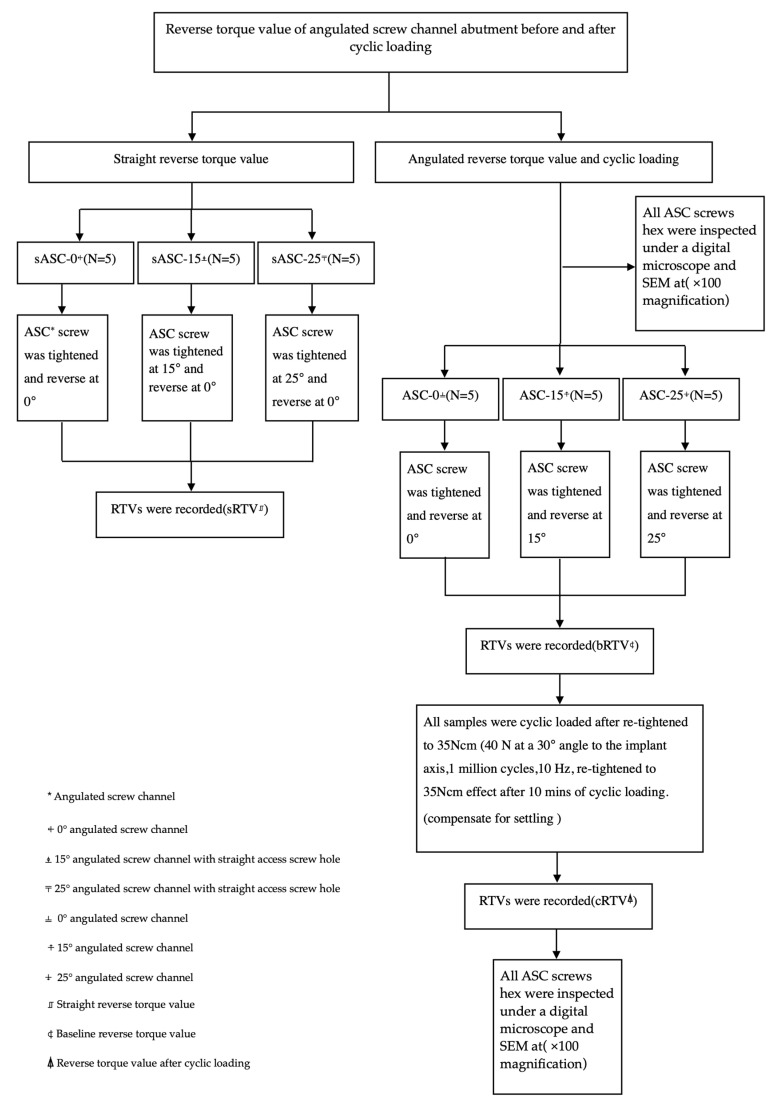
Outline of the experiments.

**Figure 6 jfb-14-00124-f006:**
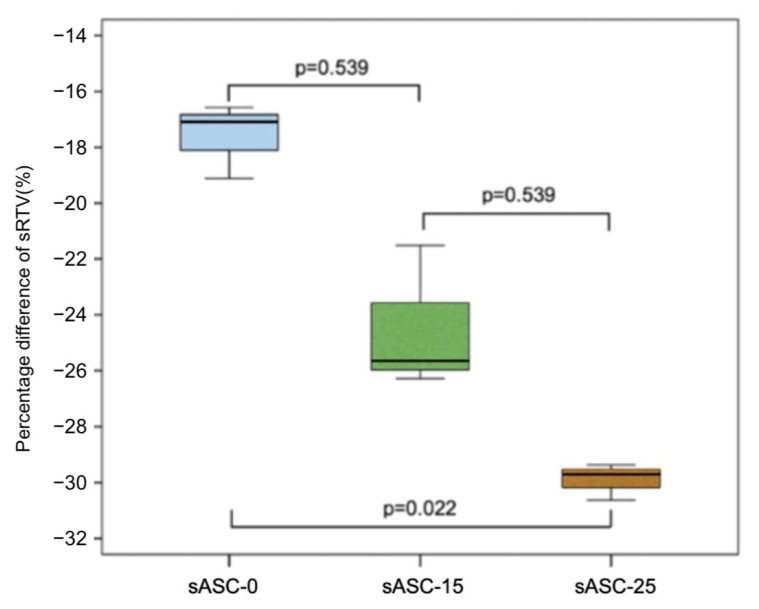
Percentage difference of sRTV among groups.

**Figure 7 jfb-14-00124-f007:**
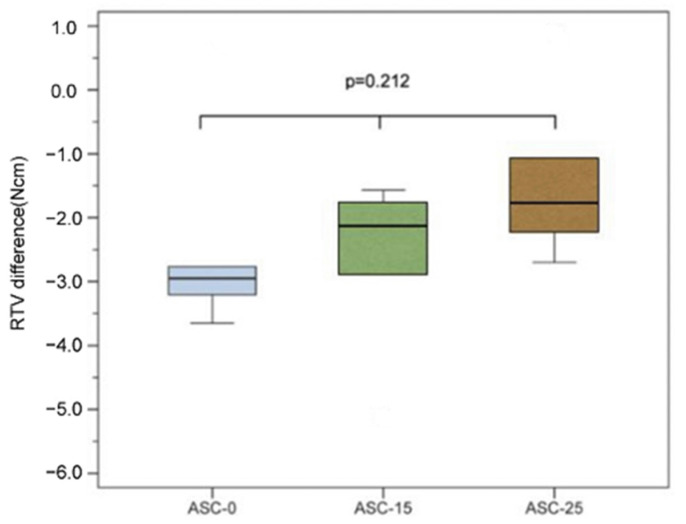
RTV difference of each group of ASC after cyclic loading.

**Figure 8 jfb-14-00124-f008:**
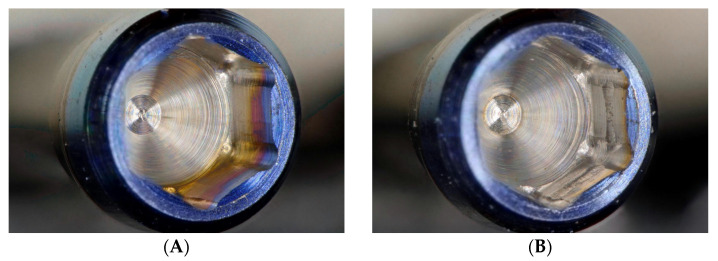
Digital micrographs with a magnification of 100×, (**A**) ASC screwhead before experiment. (**B**) ASC-0 screwhead after experiment. (**C**) ASC-15 screwhead after experiment. (**D**) ASC-25 screwhead after experiment.

**Figure 9 jfb-14-00124-f009:**
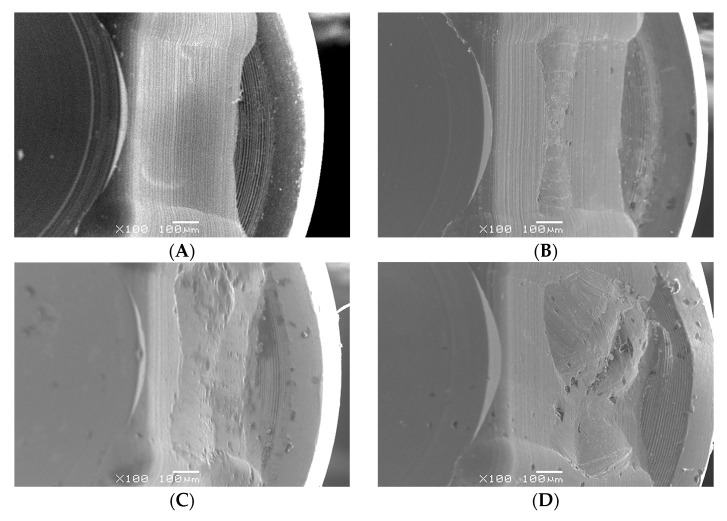
SEM micrographs with a magnification of 100×, (**A**) ASC screwhead before experiment. (**B**) ASC-0 screwhead after experiment. (**C**) ASC-15 screwhead after experiment. (**D**) ASC-25 screwhead after experiment.

**Table 1 jfb-14-00124-t001:** Results of sRTV percentage difference and RTV difference (%) after cyclic loading. (N/group = 5).

Group	Mean sRTV	Mean sRTV Percentage Difference	Median sRTV Percentage Difference
sASC-0	29.1 ± 0.43	−17.6 ± 2.6	−16.9
sASC-15	26.4 ± 0.91	−24.5 ± 2.6	−25.6
sASC-25	24.5 ± 0.23	−29.9 ± 0.6	−29.7

sASC-0, Nobel Biocare zirconia crowns with 0-degree angulated screw channels; sASC-15, Nobel Biocare special designed zirconia crowns with 15-degree angulated screw channels and straight access screw hole; sASC-25, Nobel Biocare special designed zirconia crowns with 25-degree angulated screw channels and straight access screw hole; sRTV, straight reverse torque value; RTV, reverse torque value.

**Table 2 jfb-14-00124-t002:** Results of RTV difference (Ncm) after cyclic loading. (N/group = 5).

Group	Mean bRTV	Mean RTV Difference after Cyclic Loading	Median RTV Difference after Cyclic Loading
ASC-0	28.84 ± 0.91	−2.41 ± 1.68	−2.95
ASC-15	28.87 ± 1.43	−2.70 ± 1.44	−2.13
ASC-25	27.14 ± 0.80	−1.38 ± 1.40	−1.78

ASC-0, Nobel Biocare zirconia crowns with 0-degree angulated screw channels; ASC-15, Nobel Biocare zirconia crowns with 15-degree angulated screw channels; ASC-25, Nobel Biocare zirconia crowns with 25-degree angulated screw channels; bRTV, baseline RTV; RTV, reverse torque value.

## Data Availability

Correspondence and requests for materials should be addressed to Aaron Yu-Jen Wu.

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
