# Peer review of "Reverse Torque Value of Angulated Screw Channel Abutment before and after Cyclic Loading: An In Vitro Study"

_jfb, 2023, doi:10.3390/jfb14030124_

Round 1

Reviewer 1 Report

This article is a nicely designed study, that targets area of interest of many prosthodontists in the clinical practice. Since many clinical situations still require angled abutments, predicting clinical outcome of its use is really important. Whole experimental design is done according to the reality of the clinical situation, thus making the article really worthy reading.

Author Response

Dear Reviewer,

Thank you for taking the time to review this manuscript, we appreciate your positive evaluation of our work. Those comments are valuable and important to our research. Once again, thank you very much for your warm comments. We wish good health to you, your family, and your community.

Sincerely, Yu-Hsuan Chen 

Reviewer 2 Report

The paper is interesting and well written; however, the Materials and Methods section needs more details.

Please move the references before the punctuation.

The end of the introduction can be improved, providing the motivation of the study.

Please provide more detail about the “special design of zirconia crown”.

Can you provide more information about and T-base? For instance, you can show a technical drawings or a section of the whole implant/abutment/screw/crown and so on.

Fig 2A is not sufficient in resolution, please provide a magnification and more details about the “specially designed metal jig”. Is it possible that the fixture influences the screwing? For instance, the jig clamping force should deform the implant. Please discuss this aspect.

How is the metal rod aligned with the implant and the crown?

“After 10 min, the screw was re-tightened to 35 Ncm to compensate for 137 the settling effect.” Please provide a motivation for this choice. It could be useful evaluate the removal torque after 10 min.

Why do you not consider the ISO 14801:2007 procedure for the cycling load test? For instance, you can avoid trouble related to the implant clamping. The shape of the crown can influence the results. A wide range of crown shapes lead to a wide range of results. The standardization can help in reducing variability. Please discuss this aspect and highlight the limits of the study in abstract and conclusion too.

Can you provide more detail about the cycling test protocol?

Can you provide a summary of the testing procedure at the end of the section 2?

In introduction you state that surface roughness influences the results: you should measure it before and after cycling test.

Please improve the nomenclature: for instance, you define the “mean RTV” as sRTV, but in table 1 you report “Mean sRTV”. Is it the mean of the mean?

Maybe can be useful to report the sRTV too. If you perform the statistical analysis on the sRTV instead of on the percentage, can you provide a different discussion?

Why the values in table 1 are positive and in figure 4 are negative?

The data is the same in table 1 and figure 4 so I suggest substituting table 1 with the the sRTV instead of on the percentage.

I do not understand the “Mean RTV difference after cyclic loading” in table 2: 35-28=7 .

It is not clear the meaning of the microscope examination. Interesting aspects of wear in my opinion are in the contact areas among abutment, implant and the screw. Can you provide any quantitative information?

In general, methodologically speaking, there are 3 mains weak, that must be highlighted in abstract discussions and conclusions:

- The effect of the re-tightened on the RTV is not considered,

- Different crown shapes can lead to different results. Standardized approach should be considered; for instance, this can avoid trouble related to the fixturing,

- The method is not presented with sufficient detail.

Author Response

Dear Reviewer,

Thank you for your comments, we appreciate your positive evaluation of our work. Our deepest gratitude goes to you for your careful work and thoughtful suggestions that have helped improve this paper substantially. 

Once again, thank you very much for your warm comments. We wish good health to you, your family, and your community.

Sincerely, Yu-Hsuan Chen 

Reviewer 3 Report

The aim of the paper is to understand the difference in preload acting on an abutment screw under different angles of angulated screw-retained crown and the performance after cyclic loading. The topic is up-to-date and corresponds to the journal’s area. The abstract is well structured and informative enough. The structure of the manuscript is well designed. The adequate test and statistical methods are used. The conclusions respond to the results obtained. All tables, figures and references are cited in the text. There are not missing tables, figures and references in the list.

There are a few remarks which should be edited:

1.     There are many abbreviations and it will be better if they are given in a table at the beginning of the manuscript.

2.     The aim of the study is not clearly stated at the end of the Introduction.

Author Response

Dear Reviewer,

Thank you for your precious comments and advice. Those comments are all valuable and very helpful for revising and improving our paper. We have studied comments carefully and have made correction which we hope meet with approval. Once again, thank you very much for your warm comments. We wish good health to you, your family, and your community.

Sincerely, Yu-Hsuan Chen 

Reviewer 4 Report

Dear authors, congratulations for all the effort you did to conduct this study. The topic is very interesting and appreciable from a scientific point of view and it could be a valide point to start new clinical studies. The present in-vitro study is well designed and conducted, and the manuscript is clear. There are some comments below.

The English is clear and spelled correctly but anyway a mother tongue could correct or revise it.

Abstract:

The abstract correctly summarizes the study design and purpose.

Keywords:

The keywords are correct and perfectly fitting the study design,I would add "mechanical stresses".

Introduction:

The introduction is well organized and clear. It has the right references.

LINE 44-50: When you talk about cement-retained implant prosthesis please report that other in vitro study highlighted that according the mechanical mechanical stress, the cemented-retained implant prosthesis may have better propierties compared to screwed on. It is important to cite also the mechanical aspect because it is the main factor in the early implant failure. So please cite the following article:

Cosola, S.; Toti, P.; Babetto, E.; Covani, U.; Peñarrocha-Diago, M.; Peñarrocha-Oltra, D. In-vitro fatigue and fracture performance

of three different ferrulized implant connections used in fixed prosthesis. J. Dent. Sci. 2021, 16, 397–403.

 Materials and Methods:

The methodology is well described and complete. The sample size and statistical procedures are correct.

The study has the aim to investigate the effects of the angulated screw channel abutment before and after cyclic loading and mastication forces applied to the crown and the stress distribution and the fatigue life of the implants.

I have some doubts regarding bias. For example, why did not you use a cancellous bone with a surrounding 2 mm thickness cortical bone as bone model? The implants are fixed.

Please clarify the methods and explain the bias in the discussion. You should admit it.

Results

The results are clear and supported by an adequate number of figures (suggested with colour) and tables. The statistical analysis is correct.

Discussion and Conclusions

In the limitation of the study you should discuss about my conserns reported before.

I agree with the discussion and conclusion but please, add other references of mechanical in vitro studies.

Author Response

Dear Reviewer,

Thank you for your comments. We are very grateful to your comments for the manuscript. According to your advice, we amended the relevant part in manuscript. All of your questions were answered one by one. Once again, thank you very much for your warm comments. We wish good health to you, your family, and your community.

Sincerely, Yu-Hsuan Chen 

Round 2

Reviewer 2 Report

In my opinion, the manuscript has been improved enough.

Congratulations